# Unveiling Another Dimension: Advanced Visualization of Cancer Invasion and Metastasis via Micro-CT Imaging

**DOI:** 10.3390/cancers17071139

**Published:** 2025-03-28

**Authors:** Sergey Tkachev, Vladimir Brosalov, Oleg Kit, Alexey Maksimov, Anna Goncharova, Evgeniy Sadyrin, Alexandra Dalina, Elena Popova, Anton Osipenko, Mark Voloshin, Nikolay Karnaukhov, Peter Timashev

**Affiliations:** 1Institute for Regenerative Medicine, Sechenov University, 119992 Moscow, Russia; 2Medical Institute, Penza State University, 440026 Penza, Russia; 3National Medical Research Centre for Oncology, 344037 Rostov-on-Don, Russia; 4Laboratory of Mechanics of Biocompatible Materials, Don State Technical University, 344003 Rostov-on-Don, Russia; 5Center for Precision Genome Editing and Genetic Technologies for Biomedicine, Engelhardt Institute of Molecular Biology, Russian Academy of Sciences, 119334 Moscow, Russia; 6Federal Research and Clinical Center of Specialized Medical Care and Medical Technologies, 115682 Moscow, Russia; 7Department of Pharmacology, Siberian State Medical University, 634050 Tomsk, Russia; 8A.S. Loginov Moscow Clinical Scientific Center, 111123 Moscow, Russia; 9Institute of Clinical Morphology and Digital Pathology, Sechenov University, 119991 Moscow, Russia

**Keywords:** tumor biology, metastasis, xenografts, patient-derived xenograft models, micro-CT, cancer invasion, invasion patterns, tumor budding, cell migration, esophageal squamous cell carcinoma

## Abstract

Computed microtomography (micro-CT) is a non-destructive imaging technique that enables 3D visualization of tumor progression at a resolution comparable to light microscopy without the artifacts common to conventional histology. In this study, we performed contrast-enhanced micro-CT imaging of esophageal cancer specimens via a murine orthotopic xenograft model. We visualized tumor growth, invasion, and metastasis at high resolution, preserving the natural tissue microarchitecture. We highlighted the powerful application of micro-CT in capturing the complexity of tumor invasion and dissemination, uncovering tissue-remodeling strategies. Additionally, our findings suggest that collectively migrating tumor cells exhibit organized behavior reminiscent of early multicellular organisms, reflecting the adaptive capabilities of tumor cells and their potential evolutionary origins.

## 1. Introduction

Invasion and metastasis are well-known hallmarks of cancer, with metastatic disease accounting for 60% to 90% of cancer-related deaths [1,2,3,4]. These processes are orchestrated by the complex interactions between tumors and their microenvironment and depend on the intricate interplay of multiple signaling pathways, including those that regulate cytoskeletal dynamics and phenotypic changes in stromal tissue [5,6]. Invasion patterns vary among tumor types and are influenced by multiple biological and physical factors, including the spatial localization of a lesion, cell density and stiffness in a tumor and adjacent tissues, and other histological features of the affected organ [5,7,8].

Esophageal cancer, known for its aggressive invasion and high metastatic potential, causes serious complications and is the sixth most common cause of cancer-associated death worldwide [9,10,11]. Esophageal squamous cell carcinoma (ESCC) is the main histological type of esophageal cancer [12,13,14]. To date, certain histological features of cancer invasion described for ESCC, such as the infiltrative growth pattern [15], stroma areactive surface area [16], perineural invasion [17], and the presence of tumor budding and poorly differentiated clusters (PDCs) [16,18], serve as prognostic markers, e.g., for lymph node metastasis and poor overall survival [19,20].

Currently, different approaches can be used to classify and describe the types of invasion in epithelial cancers. For example, Friedl et al. described a spectrum of cancer invasion types, including single cell migration (amoeboid and mesenchymal), multicellular streaming, and expansive growth [21]. However, there is a discrepancy between this basic classification and clinicopathologic definitions of invasion features in ESCC and other gastrointestinal cancers, since conventional histological examination does not provide a complete picture of the spatial distribution of the tumor cells and produces cutting and compression artifacts [22,23]. For instance, tumor budding (TB), which is defined as the presence of dedifferentiated single tumor cells or small cell clusters at the invasive front [24,25], appears isolated from the tumor on a 2D section, while serial sections and 3D reconstruction reveal that the tumor actually is not separated from the bud [23,26]. Therefore, an understanding of tumor expansion requires a study of the intact three-dimensional architecture of tumor tissue and its microenvironment using different visualization methods [27,28].

Despite the significant progress in cancer imaging, the existing limitations of conventional imaging techniques restrict our ability to investigate three-dimensional cellular interactions during tumor invasion and metastasis. As mentioned above, conventional histological examination provides only two-dimensional images of tissues and may be associated with irreversible tissue damage during specimen preparation [29]. Recently developed contact scanners allow the acquisition of serial sectional images of tissues in paraffin blocks for reconstruction of their three-dimensional structure. However, image acquisition with such scanners can cause tissue damage [30].

Several optical techniques are used for high-resolution spatial imaging. For example, optical coherence tomography (OCT) and confocal and fluorescence microscopy are currently used for studying macular diseases and skin conditions, as well as normal and cancer cell cultures and tissues [31]. However, sample thickness is a serious limitation for optical imaging due to light scattering and absorption by hemoglobin, myoglobin, and melanin, which are present in many tissues to different extents. This issue can be partially solved by optical clearing or increasing the section thickness, which often introduces artifacts during sample processing. Therefore, developing non-invasive and non-destructive techniques for producing three-dimensional images at microscopic resolution is highly important.

Computed X-ray microtomography (micro-CT) is an imaging technique that enables 3D non-destructive tissue visualization with a minimum voxel size of 1–5 μm^3^ [32] and provides the ability to study tumors in their native state as well as their original spatial interactions with the microenvironment [33]. Micro-CT imaging of biological samples can be performed using X-ray scattering agents that enhance the visibility of soft tissues or with a phase contrast provided by a synchrotron light source. Samples can be pretreated with various contrast agents, including iodine and phosphotungstic acid [33]. Existing protocols are compatible with conventional histological techniques and can be implemented into routine tissue processing protocols [34]. Furthermore, Gersing et al. reported that hematein-based X-ray staining is compatible with histology and combined micro-CT and light microscopy images to verify the correlative identification of specific features in both imaging modalities [35].

Previously, we successfully used similar techniques of tissue staining, image acquisition, advanced analysis in the digital environment, and subsequent histological sectioning of samples for the enucleated human eye, the penile corpora cavernosa in a rabbit, the synthetic tracheal substitute, and cell-seeded collagen scaffolds [36,37,38,39].

To date, several studies have used micro-CT for the three-dimensional imaging of tumor progression. In a pilot study, Apps et al. [40] visualized tumor invasion in adamantinomatous craniopharyngioma via micro-CT along with subsequent histological and immunohistochemical staining of the same tissue sample. Zhang et al. [41] visualized whole specimens of human esophageal adenocarcinoma using synchrotron-based phase-contrast micro-CT imaging. However, in both studies, the resolution of the obtained images was relatively low. Bidola et al. [33] successfully differentiated lung tumors in genetically modified mice from adjacent normal tissue using histopathological slides as references. In a more recent paper by Neul Lee et al., tumor tissue in the lungs of mice was visualized by micro-CT after treatment with phosphotungstic acid as well as optical tomography and field-emission scanning electron microscopy (FE-SEM) [42]. The authors suggested that these methods could be used to distinguish a tumor from healthy tissue. However, the work itself did not include segmentation, gray value analysis, or detailed descriptions of tumor nodules [42].

Recently, micro-CT has been used to acquire three-dimensional images of tumor invasion in the specimens embedded in paraffin tissue blocks (whole block imaging technique, WBI) or tissue samples (whole tissue imaging, WTI) [43,44,45]. In subsequent works, the WBI and WTI techniques were improved, and machine learning for the segmentation of tumor vessels was applied [46,47]. However, the resolution and signal-to-noise ratio of the acquired images could be improved by the sample pretreatment with an X-ray scattering contrast agent. Fortunately, phase-contrast micro-CT imaging significantly increases the spatial resolution, which allows for the identification of invasion sites and microinfiltrative carcinomas in postoperative tissue samples of the breast, cervical, and thyroid tissue samples [48]. This technique can enhance diagnostic accuracy even for the examination of small biopsy samples without damaging them [48]. Furthermore, Tajbakhsh et al. performed a WBI of follicular thyroid cancer specimens using phase contrast micro-CT and precisely identified the areas of capsular and vascular invasion [49]. However, the absence of segmentation or gray value analysis in these works did not allow for a detailed description of the invasion patterns.

In the present study, we established a patient-derived orthotopic xenograft (PDOX) model of esophageal squamous cell carcinoma and performed micro-CT imaging followed by histological analysis of iodine-stained samples. We reconstructed a 3D structure of the tumor and adjacent tissues while preserving the native histological architecture and visualized well-known tumor invasion patterns, such as infiltrative growth and several types of tumor strands. We also demonstrated the peritoneal dissemination of the tumor in three dimensions and the formation of metastatic colonies. One of the possible benefits of this study achieved by micro-CT imaging is the ability to visualize the connection between invasive tumor cell clusters and the invasion front nearby, which appeared as isolated tumor buds or clusters in micrographs. Based on the obtained data, we also proposed possible pathways of tumor spread in experimental animals depending on the tissue microenvironment and the adaptive evolution of tumor cells in secondary lesions and metastases. Thus, we demonstrated the value of micro-CT as a technique to study tumor progression and invasion while identifying potential improvements to address its limitations and challenges, positioning this work as a foundation for future studies.

## 2. Materials and Methods

### 2.1. Mice and Animal Care

Female BALB/c nu/nu mice (6 weeks old, n = 6) were purchased from the Institute of Cytology and Genetics of the Siberian Branch of the Russian Academy of Sciences (ICG SB RAS, Novosibirsk, Russia). All the animal studies were conducted following the Guide for the Care and Use of Laboratory Animals [50] and were approved by the ethics committee of the National Medical Research Centre for Oncology (protocol number 10/70). Animals were housed under specific pathogen-free (SPF) conditions (25 ± 2 °C, with a 12 h light/dark cycle) and given free access to food and water. All surgical procedures, as well as euthanasia, were performed under anesthesia with xylazine and tiletamine zolazepam hydrochloride.

### 2.2. Patient-Derived Tumor Collection

Primary tumor samples were obtained from a 60-year-old Caucasian male with histopathologically proven moderately differentiated squamous cell carcinoma of the gastroesophageal junction (stage pT3N2M0) following Garlock’s resection. Informed consent was obtained from the patient. The patient had ulcerative cancer at the lower third of the esophagus with a transition to a cardiac section of the stomach, as revealed by esophagogastroscopy.

### 2.3. Establishment of the Orthotopic Patient-Derived Xenograft Model

Patient tumor samples were transplanted into the lower third of the murine esophagus to reproduce the clinical presentation of the disease. To develop a tumor model, we performed orthotopic transplantation of a tumor fragment obtained from the previous generation of a patient-derived xenograft model into the pre-damaged esophageal wall of the lower third esophagus of immunodeficient mice. The esophagus was incised 3 mm from the gastroesophageal junction and subsequently sutured with the ligature on which the tumor fragment was located. The xenografts were passaged for three generations in the same manner. Tumor samples were collected from the third passage of the xenografts. Passaging of the animals before this study was necessary to preserve the tumor tissue and accumulate biomass (Appendix A). Animals were euthanized under anesthesia after showing signs of dysphagia and weight loss.

### 2.4. Sample Preparation

Organ samples, hereinafter referred to as specimen 1 and specimen 2 (Appendix A), were grossly observed and collected immediately after the animals were euthanized. All samples were subjected to iodine staining as previously described by Silva et al. [34]. This procedure included fixation in 10% buffered formalin for 48 h and dehydration in serial dilutions of ethanol in water (50%, 70%, 80%, 90%, 96%, and 100% by volume) for 1 h, with 1 h between each step. After dehydration, the samples were incubated in absolute ethanol with 1% iodine for 14 h. Finally, the samples were soaked in absolute ethanol, placed in plastic containers filled with absolute ethanol, and then stored at 5 °C.

### 2.5. Micro-CT Imaging

Micro-CT scanning was performed via an Zeiss Xradia Versa 520 Micro-CT 3D X-ray Microscope (Carl Zeiss AG, Oberkochen, Germany). All specimens were mounted on the sample holder in plastic tubes filled with absolute ethanol. We used different acquisition parameters for different parts of the samples (listed in Appendix A). Each sample was rotated 360°, and a source filter LE2 was used. X-ray projections were reconstructed via XRMReconstructor 12.0.8086.19558 software (Carl Zeiss AG, Oberkochen, Germany) with manually adjusted center shift values, σ = 0.5 Gauss blurring filter, and BH = 0.05 standard beam hardening correction. For drift correction, advanced motion compensation (AMC) was used. A similar approach was used in our previous studies on the same equipment [36,37].

### 2.6. Micro-CT Data Analysis

After data acquisition, the images were exported in DICOM format for volume rendering in VGSTUDIO MAX 3.4 software (Volume Graphics GmbH, Heidelberg, Germany). We used color coding based on gray values (intensity values) to visualize tumor growth in the overview and interior scans (Appendix A). Potential regions of interest (ROIs), such as the areas of metastasis and invasion into the surrounding tissue, were analyzed via the semi-automatic procedure “gray value analysis” and manual color assignment to a specific interval of gray values. These specific regions of interest were later scanned at high resolution for further analysis. To create the figures for this study, we applied the same color-coding techniques, as they were sufficient to visualize both the invasion and alterations in the tissue architecture. We performed the segmentation of the tumor tissue via automatic and semi-automatic VGSTUDIO MAX instruments on the basis of the selection of gray value intervals, such as “draw” and “region growing”. Initially, we created an ROI using the “Region growing” tool and then adjusted it using the “Draw” tool to select voxels that were not initially included. This approach allows for a more critical distinction of the tumor from adjacent healthy tissue for further analysis of the selected ROI (Appendix A). In some cases, we manually adjusted the tissue opacity to identify and highlight the tumor lesions, making certain parts of the samples more transparent than others. This approach was used to prevent color overlap in adjacent regions and to better distinguish the tumor tissue. For 3D rendering, we utilized both hardware rendering and volume rendering (Phong and Scatter) algorithms. A similar approach was used in our previous studies with the same software [36,37]. Some images were also obtained via 3D Slicer 5.8.1 open-source software for 2D analysis.

### 2.7. Histopathological Analysis

After micro-CT imaging, samples were cut according to the micro-CT projections and embedded in paraffin for further hematoxylin and eosin staining. Slices obtained from the paraffin blocks were subsequently imaged via transmitted light microscopy and scanned via an Aperio CS2 histological slide scanner (Leica, Wetzlar, Germany). Images were analyzed via QuPath 0.5.1. [51] and FIJI 2.16.0 [52] open-source software.

## 3. Results

### 3.1. Method Validation

To demonstrate the suitability of the iodine sample preparation protocol and image analysis for the visualization of tumor invasion patterns and metastatic lesions, we performed micro-CT imaging and a subsequent histological examination of patient-derived orthotopic xenografts (PDOX tumors) and adjacent tissues (Figure 1, Figure 2, Figure 3, Figure 4, Figure 5, Figure 6, Figure 7, Figure 8, Figure 9, Figure 10 and Figure 11; Appendix A). The PDOX tumors exhibited the highly aggressive phenotype with the ability to invade adjacent tissues, especially in the vicinity of the implantation site. The difference in the intensity values (which serve as a measure of X-ray scattering resulting in gray values) between the growing tumor and healthy tissues allowed us to visualize tumor invasion into the esophagus, stomach, adjacent liver, pancreas, and spleen, as well as peritoneal dissemination. We identified multiple invasion patterns, such as expansive and invasive growth, and various types of multicellular streaming, including tumor strands, finger-like projections, and the diffuse infiltration of tumor cells. Moreover, we attempted to reconstruct events during cancer progression in our PDOX model and trace the possible origin of invasion.

While describing the invasion patterns, we drew upon the works of Peter Friedl and colleagues, who proposed the terms “multicellular streaming”, “strands” (including short thick protrusions and slender tendril-like extensions), and “expansive growth” [5]. The term “finger-like projections” to describe short, thick strands invading tissues was also adopted from the literature [53]. For the classification of tumor cell clusters, we followed the histological criteria established by the International Tumor Budding Consensus Conference (ITBCC), 2016 [24]. According to this classification, clusters of up to four cells are defined as tumor buds, while clusters of five or more cells are classified as poorly differentiated clusters (PDCs) [24].

### 3.2. Tumor Invasion into the Esophagus and Stomach

Processed micro-CT images showed the infiltration and substitution of esophageal tissue by the tumor (Figure 1A,B). The invasion of the PDOX tumor into the periesophageal connective tissue spaces in the substitution area demonstrated an expansive growth pattern. Various invasion patterns could be detected at the site of tumor interaction with the *muscularis propria*. Single oval tumor cells and rounded clusters of tumor cells were present between striated muscle fibers in the superficial longitudinal layer, representing the well-known pattern of tumor budding and PDC formation (Figure 1B). Some elongated tumor cells formed thin solid strands between the layers of the esophageal *muscularis propria* and between the smooth muscle fibers of the cardiac sphincter (Figure 1A). These structures were clearly visible in the micro-CT images due to the difference in the intensity values of the tumor and muscle tissues.

**Figure 1 cancers-17-01139-f001:**
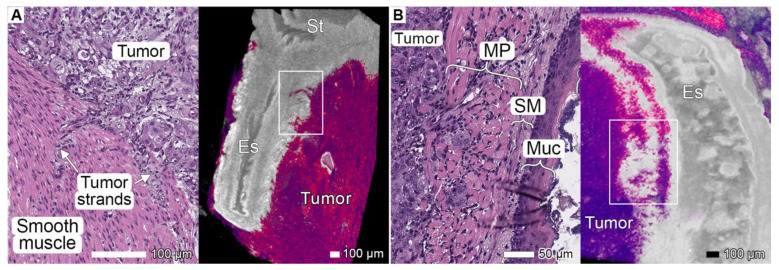
Invasion into the esophagus. (**A**) The volume-rendered model of the gastroesophageal junction near the xenograft implantation site of specimen 1, a sagittal section. Segmentation on the basis of gray values revealed infiltration into the esophageal tissue. The light micrograph of the highlighted area shows the same region of the esophagus with elongated tumor strands inside the striated muscle tissue. Scale bar, 100 μm. (**B**) Volume-rendered model of the gastroesophageal junction near the xenograft implantation site in specimen 2, a sagittal section. In both images, the tumor is identified on the basis of gray values (with the semiautomatic “region growing” instrument), allowing for the differentiation of multiple invasion areas. Both figures show signs of invasion into adjacent organs in close proximity to the primary tumor, such as the stomach, spleen, and liver. Scale bar, 100 μm. St, stomach; Es, esophagus; MP, muscularis propria; SM, submucosa; Muc, mucosa.

The tumor infiltrated the superficial layers of the gastrointestinal wall in the contact area, as shown by the differences in gray values in the micro-CT images, and, similarly, the histological examination revealed cancer cells in the corresponding projections (Figure 2). Many connective tissue spaces between the stomach and adjacent parts of the spleen and pancreas were occupied by the tumor tissue due to PDOX expansive growth (Figure 3, Figure 4 and Figure 5; Appendix A). We observed similar invasion patterns both in the glandular and non-glandular parts of the stomach. The layers of the stomach on the visceral surface were infiltrated by tumor cells, and a large tumor mass occupied the space between the superficial and deeper smooth muscle layers of the *tunica muscularis,* forming layers of tumor tissue and clusters of tumor cells among the smooth muscle cells. This disorganized structure reduced the visibility of the smooth muscle layers, which could be seen as groups of smooth muscle cells within the tumor mass (Figure 2D; Appendix A).

The *tunica muscularis* area was infiltrated almost exclusively by the oval clusters of tumor cells (tumor PDC) (Figure 2A,B; Appendix A). In the *submucosa* infiltration, we observed the elongated strands protruding through the loose connective tissue (Figure 2C,D; Appendix A), whereas the *mucosa* layers, including the *muscularis mucosae,* appeared to be uninvolved. The parietal part of the stomach wall was less affected by tumor invasion compared to the visceral part. At the same time, we observed the tumor invade into the serous coat (*tunica serosa*) of the stomach (Figure 2E) and spread through the *omentum majus* (Figure 2F; Appendix A).

**Figure 2 cancers-17-01139-f002:**
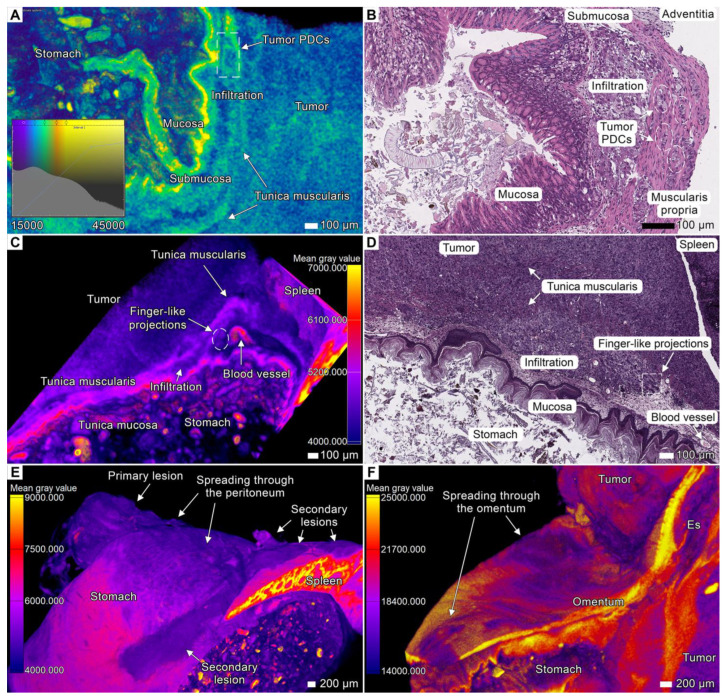
Invasion into the upper and lower parts of the stomach. (**A**) Micro-CT image showing tumor infiltration into the muscular layer of the gastrointestinal wall in specimen 1. The parula color scheme was manually applied based on gray value intervals to visualize the stratification and replacement of the stomach tissue. Scale bar, 100 μm. (**B**) Corresponding light micrograph showing the same region of the stomach. Scale bar, 100 μm. (**C**) Micro-CT image with applied gray value analysis showing invasion into the stomach wall and spleen in specimen 2. The growing tumor is in close contact with healthy tissues, leading to invasion and alterations in the adjacent tissues. The intensity values of these alterations are distinct from the intensity values of healthy tissue. The anatomical proximity in the mouse abdominal cavity shows that the tumor infiltrates the stomach as well as the spleen during this process. Gray value analysis was applied to the volume-rendered model. Scale bar, 100 μm. (**D**) Corresponding light micrograph showing the altered stomach wall adjacent to the tumor. Scale bar, 100 μm. (**E**) Infiltration into the serosa layer of the stomach in specimen 2 and invasion into the spleen through the peritoneum (carcinomatosis). Scale bar, 200 μm. (**F**) Invasion into the *omentum majus* of the stomach in the cardia region and subsequent dissemination through the omental tissue. Gray value analysis was applied to the volume-rendered model. Scale bar, 200 μm.

### 3.3. Tumor Invasion into the Spleen

In both specimens (especially in specimen 2), we observed multiple invasion and metastatic sites in the spleen. The most common type of invasion into the visceral surface was finger-like projections, the short thick strands with a rounded apex growing from the splenic hilum and infiltrating the pulp and the spaces of adjacent connective tissue (Figure 3C,D). In this region, the tumor was distributed over the parietal surface of the spleen from the ventral extremity and nearby parts of the margins. We observed clusters of tumor cells on the parietal surface of the spleen. In specimen 2, the tumor spread into the spleen via a peritoneal route (carcinomatosis) from a lesion in the tunica serosa of the stomach and formed massive lesions in the tunica serosa of the spleen and the splenic capsule, leading to infiltration in the underlying pulp (Figure 2E and Figure 3A,B). We also observed a solitary metastatic lesion in the dorsal extremity of the spleen (Appendix A). This lesion was not connected with the primary node or other lesions in the spleen, indicating that peritoneal seeding is a possible mechanism of its development.

**Figure 3 cancers-17-01139-f003:**
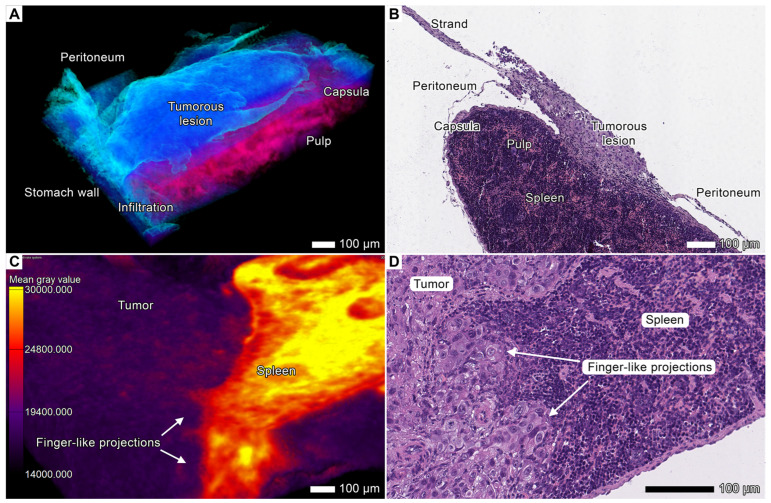
Invasion and metastasis in the spleen. (**A**) Micro-CT image showing peritoneal carcinomatosis and tumor invasion into the splenic pulp. The primary tumor lesion near the esophagus grows rapidly and spreads to more distant regions, such as the spleen. The tumor cells travel through the peritoneum and metastasize from the primary node to the superficial layers of the spleen, where they form interconnected metastatic colonies. A custom color scheme was applied. Scale bar, 100 μm. (**B**) The corresponding light micrograph of the same section in specimen 2 shows tumor invasion into the pulp, primary node, and involved peritoneum. Scale bar, 100 μm. (**C**) Volume-rendered model of the spleen showing advanced contact invasion and finger-like projection formation. Scale bar, 100 μm. (**D**) Corresponding light micrograph of the same region of the spleen with clearly distinguishable finger-like projections. Scale bar, 100 μm.

### 3.4. Tumor Invasion into the Pancreas

In both specimens, we discovered signs of massive tumorous infiltration and tumorous finger-like projections that invaded the pancreatic lobules from retroperitoneal peripancreatic connective tissue spaces. The tumor aggressively infiltrated the pancreas, replacing the original parenchyma and enveloping the remaining pancreatic tissue within the tumor parenchyma (Figure 4A–D; Appendix A). In the contact area, the acinar glandular tissue was atrophic due to compression. However, similar to the original pancreatic parenchyma, the tumor parenchyma was enveloped by acinar stromal walls, which became more compact and denser than the normal stroma. In the micro-CT images, these observations were reflected by an increase in the intensity values of the stromal fibers.

**Figure 4 cancers-17-01139-f004:**
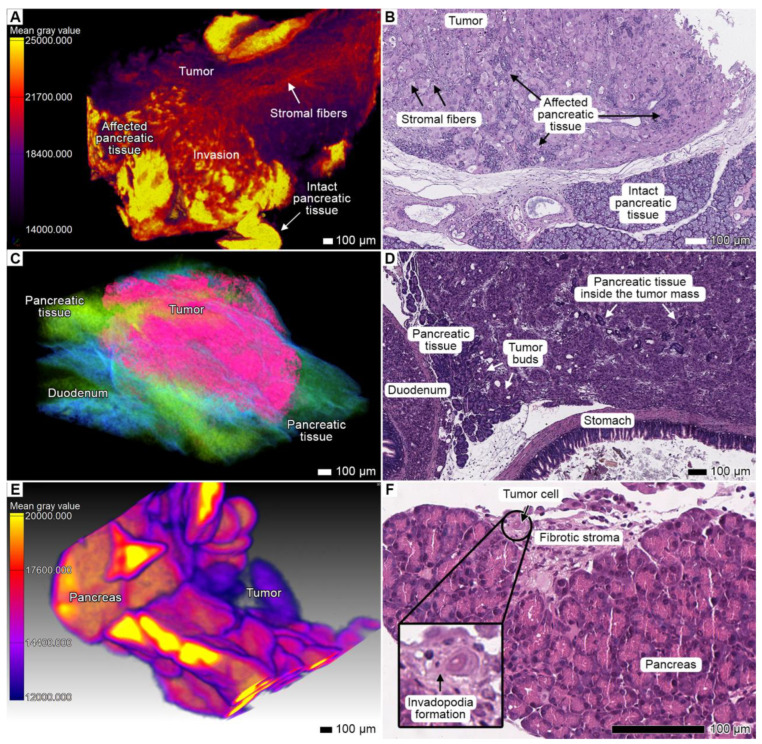
Invasion into the pancreas. (**A**) Volume-rendered model of the affected pancreas from specimen 1 with applied gray value analysis. Scale bar, 100 μm. (**B**) The corresponding light micrograph shows the same region of the pancreas. Scale bar, 100 μm. (**C**) The volume-rendered model of the pancreas with severe infiltration from specimen 2. The tumor replaced nearly all the pancreatic parenchyma; however, the shape of the organ remained the same. The tumor was segmented with the “region growing” semiautomatic instrument. Scale bar, 100 μm. (**D**) Light micrograph of this area showing the pancreatic parenchyma with the tumor tissue. Scale bar, 100 μm. (**E**) Clusters of tumor cells invading the pancreatic lobe. Gray value analysis was applied to the volume-rendered model. Scale bar, 100 μm. (**F**) Light micrograph of this region showing clearly visible invasion and cells with protrusions (presumably invadopodia). Scale bar, 100 μm.

In some less affected areas (peripheral lobules), we observed short strands with single (rarely) or multiple (more common) leading polygonal cells with spiculated cellular protrusions, likely invadopodia (Figure 4E,F). These strands invaded the lobule from the peritoneal surface. However, after the tumor sprouted through the pancreatic lobule into the peritoneum, it developed a massive node due to invasion and subsequent expansive growth of the secondary tumor lesion into the peritoneum (Figure 5 and Figure 8A; Appendix A). Apparently, this secondary nodule enveloped a part of the pancreatic lobules during its growth, which was visible in micro-CT images and corresponding micrographs (Figure 8A). 

In the pancreas, we observed the stromal fibers of tumor origin formed a continuous network spreading throughout the entire tumor mass. The resulting tumor nodes looked like reticular areas with elongated tumor clusters oriented radially to the lobule periphery between thin stromal walls (Figure 4A,B; Appendix A). These fibers likely arose from the pancreatic stroma that remained after parenchyma had been destructed by the tumor.

**Figure 5 cancers-17-01139-f005:**
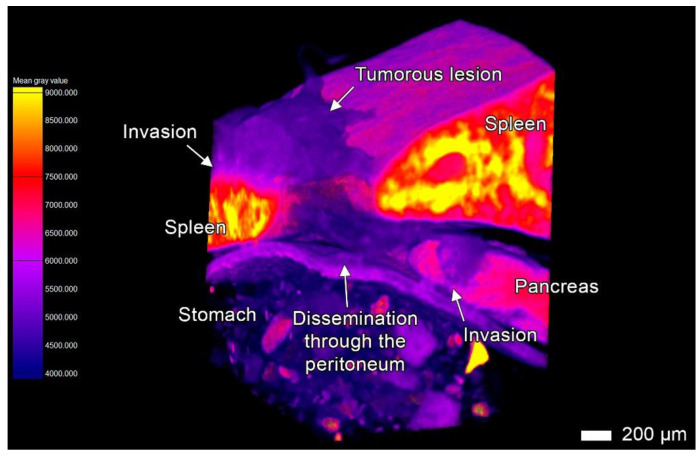
Transperitoneal invasion into the pancreas. This volume-rendered model illustrates the anatomical region between the spleen, stomach, and pancreas, highlighting the tumor progression. A tumor nodule is visible on the surface of the splenic capsule, formed by migrating cells through the peritoneum. This nodule not only infiltrated the splenic parenchyma but also spread across nearby peritoneal regions. Using the peritoneum as a conduit, tumor cells penetrated a pancreatic lobule situated beneath the spleen, independent of the primary transplanted PDOX. This invasion into the pancreas and surrounding peritoneum subsequently led to the formation of large tumor nodules, characterized by an expansive growth pattern. Scale bar, 200 μm.

### 3.5. Tumor Invasion into the Liver

In both animals, the tumor invaded the liver and formed a clearly detectable invasion front, as observed in the micro-CT images and light micrographs. The leading edge of the tumor invasion front was separated into finger-like projections (Figure 6) and rounded cell clusters (tumor buds and PDCs), with atrophic hepatic parenchyma between them and an expanding mass of tumor tissue behind (the expansive growth pattern). These cell clusters were typical of the entire invasion front, which stretched from the hilar region of the liver to the peritoneal surface, and were observed in both animals. This implies a connection between this phenomenon and the liver structure, which itself guided the invasion direction through this organ. A custom color scheme with lower transparency of the hepatic parenchyma and a color map overlay (Figure 6A) allowed for more clear differentiation of tumor finger-like projections connected to the invasion front in micro-CT images. Further histological examination of corresponding projections allowed us to conclude that the detected tumor clusters were actually not isolated nests, but rather the extensions of the invasion front (Figure 6B,D).

**Figure 6 cancers-17-01139-f006:**
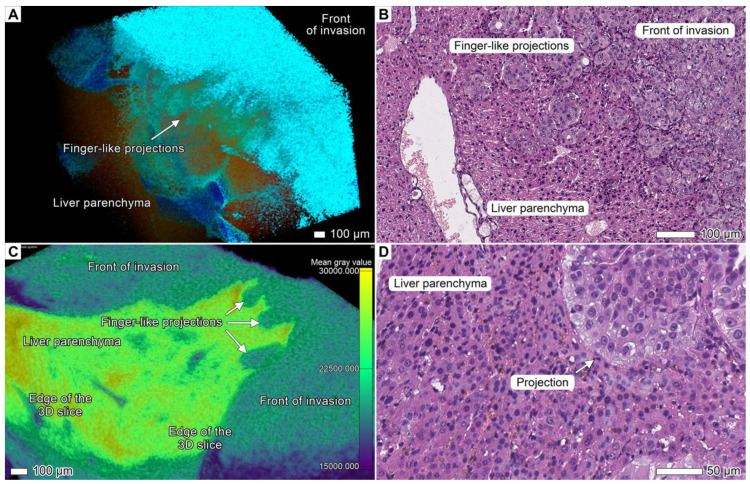
Invasion into the liver. (**A**) Volume-rendered model of the liver tissue in close contact with the tumor mass in specimen 2. The tumor invades the liver parenchyma, forming numerous finger-like projections known as buds following the invasion front. A custom color scheme was used. Scale bar, 100 μm. (**B**) Light micrograph of the same region. Tumor buds appear as isolated clusters of cancer cells that are not connected to the following invasion front. Scale bar, 100 μm. (**C**) Volume-rendered model of the liver tissue in close contact with the tumor mass in specimen 1. The tumor invaded in a similar manner as in specimen 2, forming finger-like projections followed by the invasion front. Gray value analysis was applied to the volume-rendered model with the “Viridis” color scheme. (**D**) Light micrograph of the same region with one of the tumor finger-like projections.

In specimen 2, the tumor penetrated between the liver lobes or between the liver and stomach or spleen and formed tendril-like streams (the multicellular streaming pattern) that covered the entire peritoneal surface of the lobes; however, the tumor did not invade the mesothelium or liver parenchyma (Figure 7). Similar to the finger-like projections previously observed during invasion into the organ parenchyma, tumor cells assembled into elongated strands that migrated along the mesothelium-lined surface of the liver. These strands did not invade the liver parenchyma but rather disseminated toward more distant anatomical regions in a tendril-like manner (Figure 7C–F). Some strands occupied free spaces between the liver lobes, forming growing clusters of tumor cells independently from the invasion front in the deeper layers of the organ (Figure 7D). These strands were quite difficult to detect in histological images due to the artifacts associated with sample processing or the absence of these objects in a section. Nevertheless, the identified objects that we classified as strands on micro-CT images were directly connected to the areas of tumor growth previously observed in histological images. Therefore, we considered them as variants of tumor cell dissemination in our samples.

**Figure 7 cancers-17-01139-f007:**
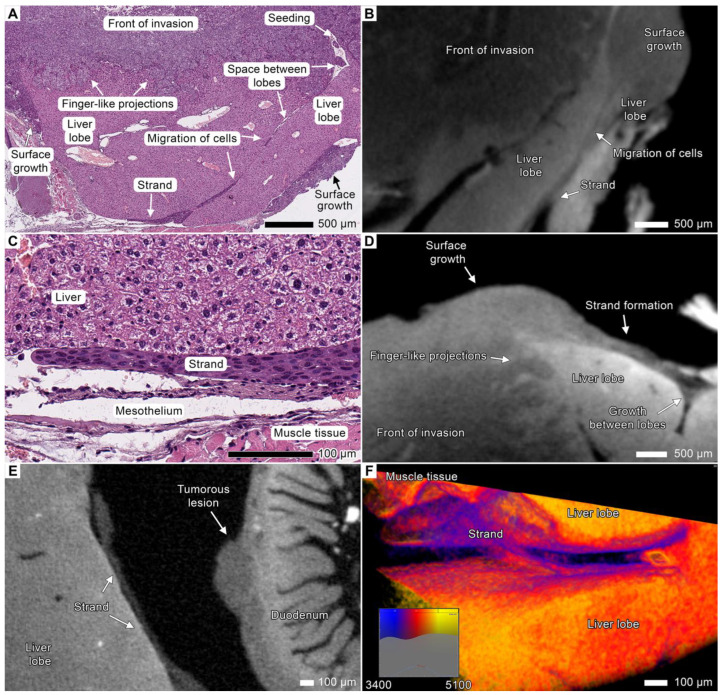
Strand formation during invasion into the liver. (**A**) Light micrograph of the liver from specimen 2 showing multiple invasion patterns. A growing tumor mass forms finger-like projections during invasive growth into the liver parenchyma. Multiple streams of tumor cells protrude into free spaces between the liver lobes; tumor cells migrate into distant sites along the liver surface as tendril-like strands. (**B**) Overview micro-CT image of the same anatomical region shows slender strands connected with a large tumor mass. (**C**) Light micrograph of a strand. (**D**) Micro-CT image of the same anatomical region from different projections showing strand formation and tumor cells in the spaces between the liver lobes. (**E**) Micro-CT image of the strand growing on the liver surface near the duodenum. This strand is directly connected to the large tumorous mass nearby. A tumorous lesion invading the tissues of the duodenum could result from such migration from a different site. (**F**) A volume-rendered model of a flat strand on the peritoneal surface of the liver between the lobes demonstrates adhesion to the mesothelium without detachment. The strand apex shows the edge of invasion, forming tendril-like streams.

### 3.6. Peritoneal Dissemination

Peritoneal carcinomatosis (PC) is a form of intraperitoneal cancer dissemination that does not arise from the peritoneum. We observed multiple disseminations through the peritoneum and abdominal serosa in both animals. Upon dissemination, tumor cells utilized these structures to spread to distant sites and to form the noduli inside the folds and ligaments of the peritoneum (Figure 5). These nodules appeared as round-shaped bulk areas of tumor cells inside the thin serous linings, as observed via both micro-CT and light microscopy (Figure 8A,B; Appendix A).

**Figure 8 cancers-17-01139-f008:**
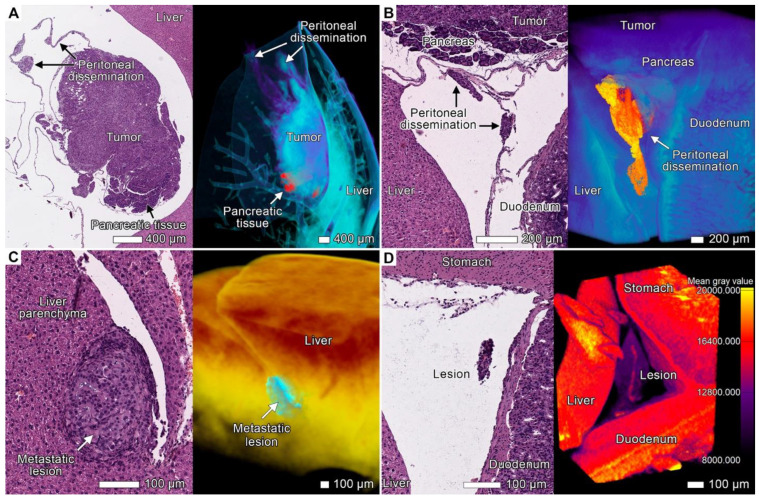
Peritoneal dissemination. (**A**) Light micrograph and the volume-rendered model of the large tumor node located between the liver lobes. The tumor tissue envelops the pancreatic tissue (appearing red in the micro-CT image) and spreads through the peritoneum as round-shaped clusters of cancer cells. The pancreatic tissue was segmented with the “region growing” semi-automatic instrument. Scale bar, 400 μm. (**B**) Light micrograph and a volume-rendered model of a cluster of cancer cells inside the peritoneal ligament between the liver, duodenum, and pancreas with massive tumor invasion. The tumor cells detached from the lesion and moved along the peritoneum, forming a round-shaped cluster inside the peritoneal ligament. The tumor and affected peritoneum were segmented with the “region growing” semiautomatic instrument. The peritoneum folds and ligaments are typically displaced or ruptured during conventional histological sample preparation, whereas micro-CT imaging allows the visualization of these structures in three dimensions. Scale bar, 200 μm. (**C**) Light micrograph and a volume-rendered model of a solitary metastasis between the liver lobes from the sample. The tumor tissue (blue) was discerned using “region growing”. Scale bar, 100 μm. (**D**) Light micrograph and a volume-rendered model of perivascular tumor invasion into the peritoneum. Both images show the peritoneum between the liver and duodenum with the presence of a small cluster of tumorous cells. A blood vessel with perivascular growth of the tumor into this part of the peritoneum can be observed. The micro-CT image allowed us to obtain more information about the spatial arrangement of native histological architecture. Scale bar, 100 μm.

The tumor sprouted into the peritoneum between the stomach, the liver, or the spleen. The mesothelial layer in this area was not visible, and the tumor border formed rounded cell clusters protruding into the peritoneal cavity.

On histological examination, we identified two types of tumor lesions in the affected peritoneum. Tumor lesions of the first type comprised flattened tumor sheets and strands in the form of tendrils overlying mesothelial cells in the absence of shedding (Figure 2F and Figure 7C–F). Lesions of the second type consisted of single rounded tumor cells or cell clusters that infiltrated the peritoneum, generally around blood vessels, and formed nodes of different sizes (Figure 8A,B,D and Figure 9A; Appendix A). Lesions of the first type were common for the visceral peritoneum on the organ surface, sometimes leading to the infiltration of adjacent tissues, whereas the second type was typically observed in free peritoneal folds.

Tumor cells use these existing connections within the peritoneal cavity to spread into the abdominal organs, and indeed, we observed several areas of tumor invasion from the affected peritoneum in both animals. The tumor invaded the parietal surface of the spleen and pancreas through the infiltrated folds and ligaments of the peritoneum (Figure 3A,B, Figure 4E,F, Figure 5, Figure 8A,B and Figure 9A) spreading through the *omentum majus* of the stomach (Figure 2F; Appendix A), the large fold of the peritoneum. The anatomical details of the tumor nodes in the visceral peritoneum of these organs are described in the corresponding sections.

During examination of the abdominal cavity, we also found tumor lesions separated both from the main invasion front and from the affected peritoneum (Figure 8C,D; Appendix A). These lesions likely originated from the peritoneal metastases spreading through the flow of peritoneal fluid. We suggest that these metastases arose from the clusters of tumor cells in the abdominal cavity formed during the expansive growth of the main tumor or large metastatic nodes (Appendix A).

**Figure 9 cancers-17-01139-f009:**
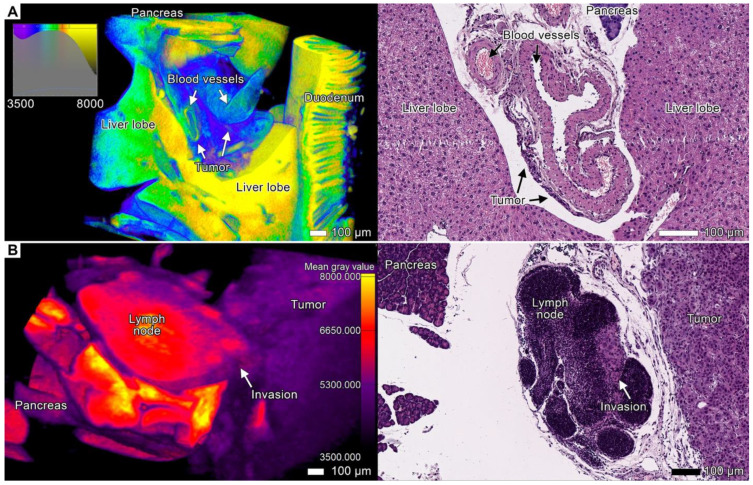
Perivascular invasion and invasion into the lymph node. (**A**) Volume-rendered model and a light micrograph of perivascular tumor invasion into the liver hilum. The left image shows 3D rendering of the area of the liver hilum, and the right image shows a micrograph of the same site. Both images demonstrate small clusters of tumor cells in the connective tissue that surrounds blood vessels in the area of the liver hilum. We manually applied color coding to the three-dimensional image on the basis of gray values. The tumor appeared as a violet and dark blue mass near the light blue blood vessel. Scale bar, 100 μm. (**B**) Volume-rendered model and light micrograph of the pancreaticoduodenal lymph node in *specimen* 2 affected by the tumor. The left image shows 3D rendering of the lymph node, and the right image shows a light micrograph of the same site. Scale bar, 100 μm.

### 3.7. Stromal Connective Tissues Define Patterns of Invasion

Micro-CT analysis showed that in parenchymal organs such as the pancreas, liver, and spleen, the stromal connective tissue fibers remain intact during tumor spread, unlike the parenchyma, which is completely replaced by tumor tissue. Of note, the tumor grew within the compartments formed by the organ stroma, without disrupting the structural fibers (Figure 4A–D and Figure 10A,B). The stroma is denser than both the tumor cells and the organ parenchyma, as indicated by higher signal intensity values in the micro-CT images (Figure 4A and Figure 10A). As the stroma was not entirely destroyed by the tumor, it formed a lattice-like structure of tumor cells separated by connective tissue fibers.

The tumor was denser than the esophageal adventitial membrane and adipose tissue at the gastroesophageal junction, which is the PDOX transplantation site. The lower density of the adjacent tissue favored the active colonization of the adventitia and adipose tissue by the growing tumor, which used available spaces and junctions (Figure 2C,D and Figure 10C,D). Nevertheless, the adventitia and adipose tissue were not fully destroyed during tumor expansion as tumor cells became entangled with the adventitial fibers, as seen both in histological and micro-CT images (Figure 10C,D).

We suggest that the connective tissues of the organ stroma and adventitia are essential for certain invasion patterns, such as finger-like projections and tumor budding. These patterns may arise from the migration of tumor cells along the interwoven stromal fibers observed both in micro-CT and histological images. In our previous work, we also observed that NIH 3T3 fibroblast cells formed interwoven projections, spreading through the existing spaces formed by the collagen fibers in seeded scaffolds [39]. Along with chemokines, collective cell migration, and other factors, the physical structure of the connective tissue may contribute to the development of these invasion patterns and tumor expansion through the fibrous lattice.

**Figure 10 cancers-17-01139-f010:**
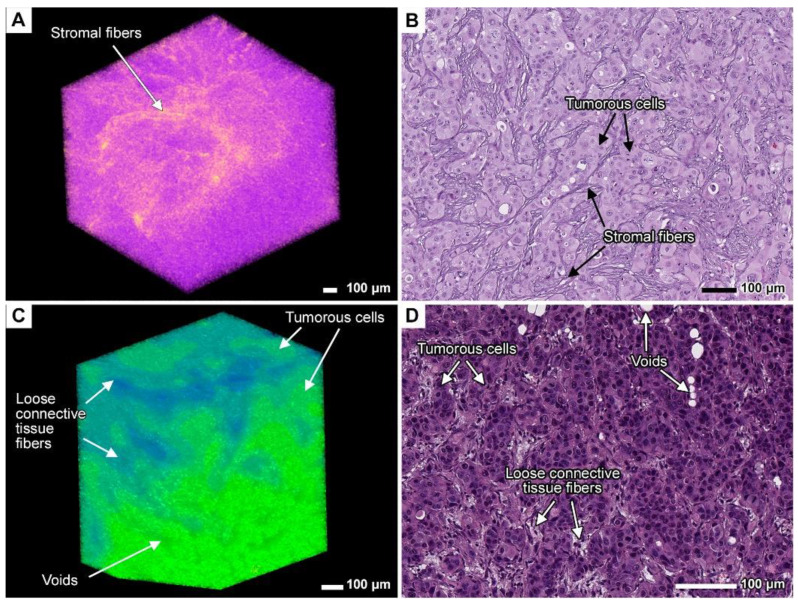
Interaction between tumor and stromal tissue. (**A**) Volume-rendered model of the pancreatic stroma enveloped by tumorous cells in specimen 2. A custom color scheme was applied. Scale bar, 100 μm. (**B**) Light micrograph of the same region. Scale bar, 100 μm. (**C**) Volume-rendered model of the esophageal adventitia near the gastroesophageal junction colonized by tumorous cells in *specimen* 1. Scale bar, 100 μm. (**D**) Light micrograph of the same region. Scale bar, 100 μm.

### 3.8. Expanding the Concept of Tumor Clusters

In order to prove that the detected isolated clusters of tumor cells were, in fact, connected to the invasion front, we examined the margin between the stomach and duodenum in specimen 2. In this region, the tumor originating from the large nodule invaded the *muscularis propria* of the stomach, forming a cluster of tumor cells, which initially appeared to be isolated from the invasion front. However, micro-CT provided the evidence that this cluster was connected to the main invasion front via a channel in the ruptured *muscularis propria*, facilitating collective tumor cell migration.

Further volumetric reconstruction revealed the previously unrecognized strand-like structure extending from the primary tumor mass into the space between the *tunica mucosa* and the *tunica muscularis* (Figure 11C). This strand separates these layers of the stomach wall, creating a defined invasion pathway. This finding indicates that tumor progression in this model is not merely a passive infiltration but rather involves active mechanical remodeling of the host tissues. By displacing and separating anatomical layers, the tumor facilitates collective migration, allowing cells to advance while maintaining an organized invasive front.

Notably, the histological examination of a corresponding projection demonstrated the presence of both a tumor cell cluster and a tumor bud. However, in contrast to the micro-CT findings, histology alone could not confirm their connection to the main tumor mass. This indicates that sectioning may significantly alter tissue architecture, potentially masking key invasion features. The ability of micro-CT to reveal continuous tumor projections can be beneficial for studying the three-dimensional complexity of invasion dynamics, revealing the events obscured in two-dimensional histological sections (Figure 11D).

Using micro-CT, we carefully analyzed the anatomical region of the gastroduodenal junction and observed a small tumor strand branching from the invasion zone toward the duodenum (Figure 11E). This strand separated the *tunica mucosa* and *tunica muscularis* layers of the stomach and could serve as a source of the tumor bud detected during histologic examination (Figure 11F). Our findings suggest that the strategy of separating tissue layers and advancing through areas of least resistance (between the layers) might serve as a universal mechanism in tumor invasion. Moreover, it demonstrates that even more distant tumor buds can be connected to the invasion zone by tumor strands that are lost during conventional histological processing (Figure 11F).

**Figure 11 cancers-17-01139-f011:**
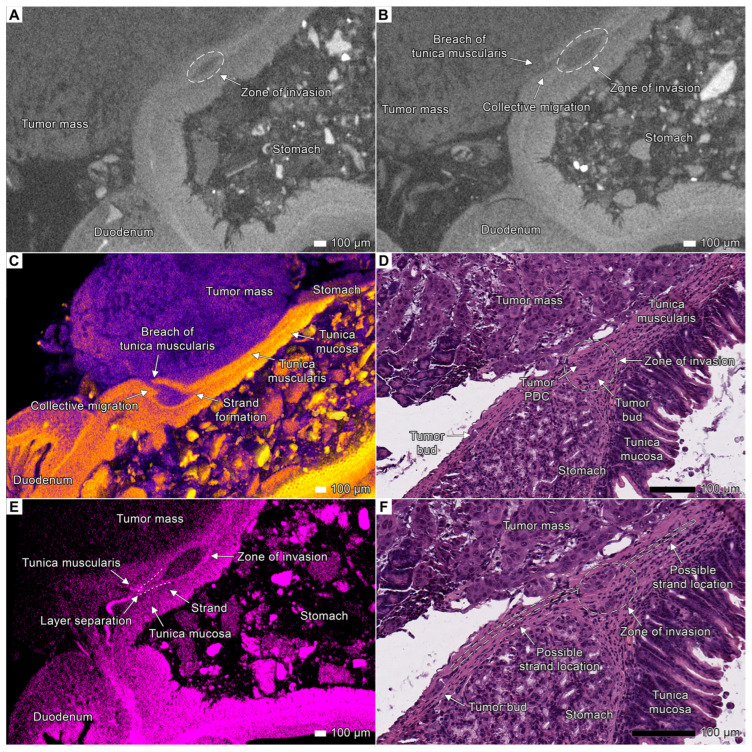
The projection plays a critical role in accurately visualizing tumor invasion. Micro-CT’s ability to non-destructively inspect objects from multiple angles allows the same area to be viewed in completely different ways. In *specimen* 2, we identified a small region at the border between the stomach and duodenum where the tumor had invaded the stomach tissue by penetrating the *tunica muscularis* to form an invasion zone. (**A**) The tumor cluster appears isolated from the nearby invasion front. (**B**) However, taking a few slices away, it becomes clear that this seemingly isolated cluster is, in fact, directly connected to the main tumor mass through a channel in the *tunica muscularis*. This channel enabled the collective migration of tumor cells. (**C**) A volume-rendered model of the invasion site. Using the region-growing tool, we segmented the tumor based on gray values and highlighted it in blue while applying a custom color scheme to the stomach and duodenal tissues. This approach enabled clear visualization of the tumor’s invasion into the stomach and the collective migration of cells from the tumor mass into the invasion zone. (**D**) A histological examination of the same invasion zone shows an isolated tumor bud and a poorly differentiated cluster (PDC). However, these structures are not truly isolated; rather, they remain connected to the main tumor mass, representing a single, continuous tumor projection. This highlights the limitations of two-dimensional histological projections in accurately portraying the three-dimensional nature of tumor invasion. (**E**) We observed that a small strand separating the layers of the *tunica muscularis* and *tunica mucosa* extended from the invasion zone toward the duodenum. We hypothesize that this strand was the source of the tumor bud visible in the micrograph. (**F**) Micrograph of the invasion zone with overlaid contours indicating the presumed location of tumor strands, as observed in the micro-CT images. Scale bar, 100 μm.

## 4. Discussion

### 4.1. Visualization of Tumor Invasion

Invasion is a key indicator that reflects the relationship between a tumor and surrounding tissues [54]. Invasion strategies such as single-cell infiltration as well as the formation of tumor clusters (buds and PDCs) and strands have been previously described [5,55,56]. However, little is known about the site-specificity of invasion patterns [57]. Visualizing the invasion front using conventional histology remains challenging, or even technically impossible, as the topology of the tumor and surrounding intact tissues is partially disrupted during tissue sectioning [22,23]. In contrast, micro-CT imaging enables the visualization of tumor spreading at a magnification comparable to that of light microscopy and also allows for reconstructing tumor progression events [29]. This approach can help us to understand how the tumor microenvironment influences emerging invasion patterns. The main advantage of micro-CT imaging is the possibility of the 3D reconstruction of the tissue architecture [29], which can be laborious and cumbersome by traditional histological examination [58,59]. Previous studies have demonstrated the aforementioned advantages of micro-CT imaging for many tumor types [33,40,60,61], including esophageal cancer [41]. Thus, micro-CT has become one of the most widely used technologies and is considered the gold standard in 3D non-destructive preclinical imaging, especially for cancer research [62,63].

### 4.2. Considerations for the Model Selection

In this study, the orthotopic tumor xenografts of esophageal squamous cell carcinoma, transplanted from a patient into immunodeficient mice, exhibited high invasive and metastatic potential. Various stages of tumor progression and microscopic features were visualized using both micro-CT and light microscopy (Appendix A). We selected PDOX models because they more accurately replicate tumor progression and metastatic spread compared to subcutaneous patient-derived xenografts. Additionally, PDOX models preserve the structure and composition of the primary tumor [64]. Experimental tumor models have advanced significantly, with xenografts providing a well-established and reproducible platform for studying tumor biology [65]. However, traditional cell culture-based xenografts often fail to capture the heterogeneity of human tumors due to clonal selection during in vitro culturing [65,66,67]. Patient-derived xenografts (PDX) overcome many of these limitations by directly transplanting tumor fragments into immunodeficient mice, preserving the histological and genetic characteristics of the original tumor [64,66,67]. Among PDX models, patient-derived orthotopic xenografts (PDOX), in which tumors are transplanted into their anatomical site of origin, retain critical tumor–microenvironment interactions and metastatic behavior, closely mimicking the parent tumor [64,66]. It is important to note that injecting tumor cells into the bloodstream or directly into common metastatic sites does not fully replicate the metastatic cascade, which involves primary tumor growth, invasion into surrounding tissues, intravasation, and the colonization of distant organs. Spontaneous metastasis models, in which subcutaneous xenografts or a significant portion of orthotopically transplanted tumor fragments and cell cultures begin to spread throughout the body, provide a more reliable representation of the full metastatic process [64]. The forefront of tumor modeling is represented by humanized mouse models, which incorporate human-like immune systems to facilitate the study of immune-tumor interactions [68,69,70]. However, their complexity and high cost currently limit their use in preclinical research [70]. In this study, we selected the PDOX model due to its ability to faithfully replicate the tumor microenvironment and interactions with surrounding tissues while maintaining practical feasibility.

### 4.3. Invasion Patterns

We identified the patterns of primary and metastatic expansive growth and various subtypes of invasive growth, including the formation of strands and finger-like projections and diffuse cell infiltration [20,21,25,71]. We observed a classical example of expansive tumor growth previously described in the literature [5], where the tumor mass pushed outward from its original site, invading the adjacent organs. In this manner, the tumor invaded the tissues of the esophagus and stomach (Figure 1 and Figure 2), as well as the nearby spleen, pancreas, and liver (Figure 3, Figure 4, Figure 5, Figure 6 and Figure 7). This behavior is typical for the orthotopic models of esophageal cancer, which also demonstrate peritoneal dissemination involving the liver, lymph nodes, and other organs [72,73].

### 4.4. Tumor Budding Revisited

The scattered clusters consisting of approximately of four cells (buds) or more than five cells (PDC) often appeared as finger-like projections of the invasion front (Figure 2C,D, Figure 3C,D, Figure 4A,B and Figure 6; Appendix A). However, the tumor cell clusters observed in the histological images were isolated from the invasion front, as the micrographs are two-dimensional. However, as can be seen in 3D images, the clusters could be the direct extensions of the main invasion front, maintaining the connection to the growing tumor mass (Figure 6A,B and Figure 11). We included a separate section in the Results section to clearly demonstrate that even the smallest tumor clusters were not actually isolated from the invasion front (Figure 11). For this purpose, we analyzed a specific area at the margin between the stomach and duodenum, where the tumor had invaded the muscularis propria. Using micro-CT, we observed that a small cluster of tumor cells was connected to the tumor mass. Similarly, the finger-like projections we detected were revealed to be direct extensions of the invasion front. However, in histological examination, these structures could appear as isolated islets due to the limitations of two-dimensional histological projections. Accumulating evidence implies that tumor buds, for instance, are associated with the invasion front in most cases [23,25]. Various techniques based on the three-dimensional analysis of histological architecture, such as micro-CT, can significantly improve our understanding of invasion.

This notion has been reported previously [26] and has been suggested in the literature [25,53]. In the recent work of Yoshizawa et al. [74], the pattern of the invasion front of cholangiocarcinoma, visualized via confocal scanning laser microscopy, was similar to the pattern we observed in the liver and other invasion sites via micro-CT (Figure 2C,D, Figure 3C,D, Figure 6A,B and Figure 11). This suggests that the tumor clusters were topographically related to the invasion front. Thus, our results obtained via micro-CT and histopathological methods confirmed that tumor buds were connected to the front of tumor invasion in line with the several recent studies.

Currently, scoring systems for TB are implemented in clinical pathology for certain carcinomas [23]. However, previous results and our findings indicate that buds and PDCs are connected to the invasive front. This type of tumor growth is highly likely to be associated with the more aggressive tumor phenotype that had already undergone EMT [53]. We assume that the quantitative assessment of tumor buds can vary greatly, since it depends on sectioning, the quality of the sample preparation, and counting by the pathologist [23,25]. We suggest that TB reflects the more complex phenomenon of tumor growth formation due to EMT, which we observed on micro-CT. Thus, the presence of even several buds can already be regarded as a marker of advanced, more malignant, invasive tumor growth and, consequently, a worse prognosis, as has been shown for ESCC [18,71].

### 4.5. Tumor Invasion as a Process of Coordinated Tissue Remodeling

Our findings suggest that tumor invasion might not be merely a process of localized tissue penetration, but rather a structured mechanical remodeling of the host tissues. In particular, we identified a strand-like structure that extended from the tumor mass separating the tunica mucosa from the *tunica muscularis*. This structural adaptation created a pre-formed invasion pathway, facilitating collective migration while maintaining tumor cohesion (Figure 11C). Unlike classical models of tumor invasion, which emphasize enzymatic extracellular matrix degradation, this observation suggests that tumors may also employ mechanical displacement strategies to advance through tissue compartments. This implies an alternative invasion mechanism, where structural disruption, rather than destruction, enables tumor cells to infiltrate new areas. Such strand-like invasion structures highlight current limitations of histology in describing the complexity of tumor invasion. Histological sections of the same region (Figure 11D) showed isolated tumor buds and clusters, which could be misinterpreted as disconnected invasion events. However, volumetric reconstruction by micro-CT revealed that these structures preserved their connection with the primary tumor via the invasion strand, providing a clear migration route. This implies that we may have overlooked tumor connectivity, as conventional two-dimensional techniques may fail to detect certain invasion patterns, particularly those involving structural displacement rather than destruction. Integrating micro-CT into tumor invasion studies would allow for a more accurate interpretation of spatial dynamics and provide new data essential for understanding metastatic progression.

The invasion mechanism we identified aligns with the theoretical framework proposed by Peter Friedl and colleagues [21,75], who described complex patterns of multicellular and single-cell tumor migration during cancer progression. While tumor invasion is often considered a process of individual cell dissemination or bulk tumor expansion, our findings suggest a distinct structural remodeling strategy. In particular, the strand-like structure we observed enabled maintaining a coordinated invasive front and displacing the host tissue, possibly facilitating collective migration. This supports the idea that tumors do not rely on a single invasion mechanism but rather employ complex strategies, adapting their migration patterns to the structural and mechanical properties of surrounding tissues.

The patterns of tumor cell migration we observed resemble the collective migration of social amoebae such as *Fonticula alba*. The way in which *Fonticula alba* changes its morphology depending on the medium (agar or bacterial mats), as shown by Toret et al. [76], resembles the adaptive phenotypic plasticity in collective migration and invasion that we observed in tumor cells. Studying social amoeba plasticity, along with leader–follower dynamics during collective migration, could help to understand how tumor cells coordinate their actions during their migration. Methods used to study social amoebae, such as *Dictyostelium*, might prove useful for investigating the signaling orchestrating tumor cell migration [77]. One could suggest, supported by recent studies and our observations, that tumor cells—once thought to be selfish—successfully cooperate to perform joint tasks [78]. Moreover, since collective migration in the form of sheets and filaments (strands) is an ancient and universal program of cell migration [76,79], we suggest that tumor cells employing these patterns may be triggering ancient atavistic programs during carcinogenesis. However, it remains unclear whether these patterns result from the reactivation of ancient programs or their reinvention as part of tumor progression and adaptation [78].

### 4.6. The Cyclical Nature of Tumor Dissemination

Based on the literature and our observations using micro-CT imaging and light microscopy, we propose the possible routes and pathways of tumor dissemination through the different anatomical areas in the experimental animals shown in Figure 12. This diagram describes possible invasion patterns, involved anatomical areas, and routes of dissemination.

**Figure 12 cancers-17-01139-f012:**
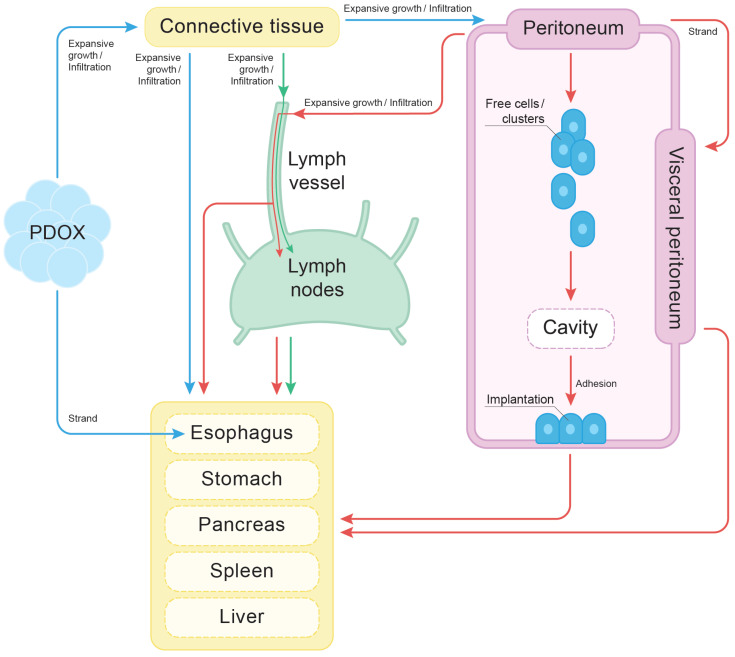
Proposed pathways of tumor dissemination and site-derived invasion patterns. The diagram illustrates the possible mechanisms of spreading of the studied PDOX tumors. The tumor invaded into the connective tissue near the implantation site via expansive and infiltrative growth; at the same time, it formed a strand that invaded denser tissues such as smooth muscle tissue (blue arrows). While invading the peritoneum, cancer cells formed strands on the mesothelial surface, while free cancer cells in the peritoneal cavity invaded other regions of the peritoneum or into the peritoneal lymph nodes (red arrows). The green arrows represent spreading via lymphatic vessels, collecting ducts, and lymphatic nodes.

Our study could provide evidence supporting the self-sustaining, cyclical nature of carcinomas that has been described previously by Friedl et al. [5]. One cause of this cyclical process is the reciprocal way in which tumor cells affect surrounding tissues and vice versa, which together drive cancer progression [5]. The PDOX tumor gave rise to secondary lesions, which in turn acted as independent sources of further tumor spreading. Each secondary cell cluster in the liver, spleen, peritoneum, and lymphatic nodes could serve as a new primary site, creating new metastatic lesions (Figure 12). Furthermore, tumor expansive growth triggered its invasion into the adjacent organs by pushing cells out. All these secondary sites, in turn, remodeled the stroma of the organ and the surrounding parenchyma that promoted the development of the tumor microenvironment [5]. For instance, finger-like projections and migrating flat strands might serve as the starting points for further metastatic spread.

### 4.7. The Finger-like Projection Type of Invasion in the Connective Tissue

Upon the start of the growth of the transplanted tumor inside the abdominal cavity, the first affected tissue after granulation was loose fibrous connective and adipose tissues surrounding the esophagus and the gastroesophageal junction (Figure 1, Figure 2 and Figure 5; Appendix A; Figure 12, blue arrows). The tumor most likely invaded into the intercellular spaces, forming the finger-like projections. Such projections could originate from the amoeboid cells that might be involved in the colonization of the connective tissue. The tumor spread into the retroperitoneal cellular space and quickly reached all the sites crucial for further spread, such as the posterior stomach wall, pancreas, hilum of the liver, and spleen (Figure 2, Figure 3, Figure 4, Figure 5, Figure 6, Figure 7 and Figure 9A; Appendix A). The finger-like projection type of invasion might be related to the presence of connective tissue accompanying the vessels in the hilum of these organs or surrounding them, allowing tumor invasion through the connective tissue into the deep layers of these organs (Appendix A). Tumor cells probably migrated from the lymphatic vessels into the lymphatic nodes in these organs (the primary stomach wall) (Figure 9B; Appendix A).

Being in contact with dense tissues, such as the organ’s parenchyma, the tumor formed finger-like projections, accompanied by proteolysis of the extracellular matrix, as shown in the corresponding micrographs (Figure 3C,D and Figure 6A,B; Appendix A). These results are in line with previous work that reported the strands in the tissues with increased density [55,80].

### 4.8. Peritoneal Dissemination

After the tumor grew through the retroperitoneal fat tissue, it spread out into the peritoneal cavity (Figure 5, Figure 7 and Figure 8; Figure 12, red arrows; Appendix A). Both orthotopic murine xenografts demonstrated peritoneal dissemination, although it is not very common in the clinic [81]. However, further studies are required to compare the invasion mechanisms of such spreading pathways, the vector of clonal evolution, and the site-specific invasion patterns between model organisms and patients [66].

In all the cases discussed above, peritoneal dissemination occurred simultaneously at multiple sites, mostly between the stomach and the adjacent liver and spleen. Thus, the peritoneal cavity could cause further tumor spreading by different pathways [82]. The tumor surface protruding into the peritoneal cavity during expansive growth could act as a source of free peritoneal cancer cells and their clusters (Appendix A). Similar behavior has been described for the tumors complicated with carcinomatosis [83,84]. Although such clusters have not yet been described for esophageal SCC, in the histological images, we often observed small pedunculated cell clusters on the tumor surface protruding into the abdominal cavity. Free tumor cells and clusters could migrate to any organ of the abdominal cavity via the flow of peritoneal fluid, adhere, and undergo trans-mesothelial metastasis [84].

In the implantation area, peritoneal metastases also formed strands with spiculated apex cells. These projections, probably invadopodia, might have served for invasion into the submesothelial tissue (Figure 4E,F) [5,85]. These strands were enriched with fibroblast-like cells, which could act as leader cells. For instance, they can lead strands by lysing the extracellular matrix with invadopodia and disrupting the desmosomes between tumor cells and adjacent normal cells [86,87]. Altering the invasion pattern might help to grow through the dense layer of the mesothelium. In general, the second route of esophageal SCC invasion is likely to involve the passage through the peritoneum, with the formation of free cells and cell clusters.

During growth on the surface of the mesothelium, the tumor forms migrating flat strands that often start at the site of ingrowth of the underlying tumor into the mesothelium (Figure 7C–F). We detected such strands at the stage of tumor cell implantation into the peritoneum but not during multicellular migration, i.e., as an invasive strand. It is different from typical strands that grow deeper into the tissue during the spreading of the tumor cell layer over the surface of the mesothelium [82].

Finally, the most frequently observed pattern of peritoneal invasion was infiltration, followed by the formation of nodes of varying sizes within the peritoneum due to the expansive growth of the secondary lesions (Figure 8A,B; Appendix A). It was associated with the invasion into the connective tissue of the peritoneum from the underlying organs and with the tumor spreading along the peritoneum. Thus, we suggest that peritoneal dissemination and infiltration within the peritoneum might occur at the final stage of the development of metastases. They may result from tumor invasion into the peritoneum, which could lead to translymphatic metastases in other parts of the peritoneum, lymph nodes, or other organs [82].

Among the observed types and routes of invasion, we did not find any examples of tumor spreading that could be explained exclusively by the hematogenous metastatic route. The solitary metastatic lesion between the liver lobes in specimen 2 most likely arose from the spread of peritoneal fluid rather than hematogenous spread (Figure 8C). We suggest that the spleen surface metastasis in specimen 2 formed through a similar mechanism, as it was not located near secondary lesions associated with peritoneal carcinomatosis (Appendix A). Previous studies have shown that distant hematogenous metastases are most likely associated with a poor prognosis and develop at a late stage of the disease [10,11]. This implies that esophageal cancer might acquire the ability for hematogenous metastasis at the final stages of its development, mostly via other routes and mechanisms of invasion, such as contact invasion, the lymphogenous metastatic route, and migration along body cavities (peritoneal carcinomatosis) and lymphatic spaces (Figure 8 and Figure 9B; Appendix A).

### 4.9. Cellular Heterogeneity and Adaptive Phenotypic Plasticity

The diversity of invasion patterns that we have observed could be the result of adaptive phenotypic plasticity, which is defined as an ability of cells to adopt various phenotypes without changing their genotype [88]. In two independent mouse models, tumor cells spreading from the original region of the orthotopically transplanted patient tumor fragment exhibited similar patterns of collective migration under comparable conditions. The coexisting diversity and consistency suggest that during collective migration, tumor cells might adopt strategies that maximize their proliferation and survival.

We suggest that the correlation between the modes of invasion of tumor cells into the liver and the acquired phenotypes (Figure 6 and Figure 7; Appendix A) could serve as an example of adaptive phenotypic plasticity [88,89]. Cells that infiltrated the dense liver parenchyma as finger-like projections displayed a phenotype of non-keratinizing large epithelial cells with moderately abundant amphiphilic cytoplasm, large hyperchromatic nuclei, and prominent nucleoli. In contrast, tumor cells that migrated as strands along the surface of the liver capsule were phenotypically distinct, forming predominantly solid structures composed of smaller epithelial cells with sparsely eosinophilic cytoplasm and medium-sized hyperchromatic nuclei. Notably, these cells originated from the same source, and we observed a gradual transition between these phenotypes within the liver, providing evidence that adaptive phenotypic plasticity could drive cellular heterogeneity.

Another example of the cell phenotype shift is the invasion into muscle tissue (Appendix A). Cells changed their phenotype from large-cell to small-cell as they invaded dense muscle fibers, moving from non-keratinizing sheets of large polygonal cells with moderately abundant eosinophilic cytoplasm, hyperchromatic and pleomorphic nuclei with nucleoli to dense non-keratinizing diffuse sheets of small cells with scant cytoplasm, hyperchromatic nuclei, and absent nucleoli.

Furthermore, tumor cells migrating through the abdominal cavity in the form of multicellular strands or sheets could later invade dense organ parenchyma and undergo further phenotypic changes, or form secondary metastatic nodules (Figure 5 and Figure 8; Appendix A). Under favorable conditions, these nodules can grow exponentially and serve as new sources of collectively migrating tumor cells. The newly disseminated cells will, in turn, adopt the phenotypes optimized for their specific microenvironment. This fact could serve as further evidence that tumor invasion is an adaptive and context-dependent process that promotes tumor cell survival and proliferation.

However, it should be noted that the expansive growth of the secondary lesions creates ischemic areas, where cells are constantly evolving and rapidly adapting to the properties of the microenvironment, resulting in the selection of the most aggressive clones [1,2]. This might trigger a variety of cell dissemination modes, as well as cellular phenotypes (Appendix A). Invading various organs and tissues results in different phenotype dissemination strategies under constant clonal selection (e.g., changing from a multicellular to a single-cellular mode of dissemination or changing the phenotype from mesenchymal to epithelial type) [53]. The interactions of the growing tumor with adjacent tissues change its microenvironment, resulting in inflammation, structural alterations, extracellular matrix remodeling, necrosis, and an altered balance of soluble signaling molecules and enzymes. This could shift the phenotype of tumor cells and facilitate tumor plasticity required for its invasion. Therefore, we suggest that the observed heterogeneity of tumor cells and invasion strategies might arise from the different properties of the extracellular matrix in the microenvironment (Figure 6, Figure 7 and Figure 11; Appendix A). This is supported by the fact that tumors from both animals showed similar invasion patterns in the corresponding regions and had similar histological features, implying that they might evolve in a similar manner due to their similar microenvironments. However, we did not always observe such diversity of invasion strategies during histological examination, which can be caused by the altered tissue architecture after the sample preparation or absent regions of interest in the section. To exclude the influence of initial intratumoral heterogeneity in the patient-derived xenograft and to distinguish between adaptive plasticity and clonal selection in the observed cellular heterogeneity, further studies using molecular phylogenetics are required. Furthermore, it is still unknown how epigenetic alterations could affect the results of clonal selection [88].

## 5. Limitations of the Study

We selected micro-CT as the primary imaging modality for this study because it enables the high-resolution tracking of tumor progression and metastases at the level of cell clusters while allowing histological validation of these regions [32,62,63,90]. By contrast, PET imaging assesses cancer progression at the whole-organ level through the direct measurement of chelator and radiotracer uptake at specific time points [91,92]. Although PET provides valuable metabolic insights and has a faster translational pathway, it does not capture detailed morphological tissue structures [91,92,93]. For example, Benfante et al. demonstrated the biodistribution of a ^64^Cu-labeled chelator in a murine model, emphasizing how radiopharmaceutical uptake patterns evolve over time [94]. We speculate that future advancements will enable a multimodal approach, combining in vivo PET imaging with ex vivo micro-CT enhanced by contrast agents and correlated with histological examination of regions of interest [62,91]. This integrative strategy would offer a more comprehensive view of tumor morphology, metabolic activity, and interactions with surrounding tissues, addressing both structural and functional aspects of tumor progression.

We used an iodine staining protocol that provides a rough visualization of only the cell mass but not of individual cells, as it is based on iodine binding to polymeric carbohydrates (specifically glycogen) in animal cells [34,95]. However, relatively few staining techniques that enable the visualization of cells and their nuclei are currently available [96,97]. We suggest using the method proposed by Müller et al. for the micro-CT imaging of cancer cells, although it is more time-consuming than iodine staining [97]. Visualization of cells and their nuclei is promising for micro-CT tumor imaging, since tumor cells can be distinguished due to the pronounced atypia of tumor cells and their nuclei. *Müller* et al. described the staining technique as well as the segmentation of different liver cells [97]. Therefore, specialized software (such as VGSTUDIO MAX 3.4) would allow the identification of individual cancer cells in a sample stained following the mentioned protocol (segmentation could be based on their density or other parameters). The use of antibody-conjugated radiolabels could provide greater specificity of imaging, allowing for the visualization of specific immune cell populations interacting with tumor cells in 3D environments. For instance, Sanna et al. demonstrated the successful functionalization of gold nanoparticles with nanobodies, referred to as Nano2, targeting specific proteins such as GFAP [98]. These conjugates significantly enhanced the specificity of imaging modalities, such as X-ray phase contrast tomography, enabling the precise identification of cell types within a 3D morphological context. This approach could be adapted to target immune cell markers and study their interactions with tumor budding cells within the tumor microenvironment. There is also a simpler protocol for visualizing nuclei on the basis of lead acetate staining [99]. However, unlike the iodine protocol, this method does not provide high-quality visualization of blood vessels, which can be critical for studying tumor neovascularization. Since the tumor stroma plays an important role in tumorigenesis and tumor invasion [1,5], protocols based on the use of phosphotungstic acid may also be useful for further studies due to its specific binding to collagen [100,101].

In our study, reconstruction was carried out in the global byte scaling mode, which means that the minimum (0) and maximum (65,535) gray values correspond to the minimum and maximum brightness values found in a particular dataset. This also means that a fixed gray value is not equivalent to the same physical density in different scans, even if the scan and reconstruction parameters (accelerating voltage, source filter, beam hardening correction factor) are identical. In further studies, it is advisable to calibrate the samples via phantoms (such as distilled water or air) or scan them under the same conditions to objectively compare the resulting changes in the density of normal and tumor tissues [36,37]. The voxel size provided by a lab-based X-ray machine (such as Zeiss Versa 520) that allows the desired signal-to-noise ratio to be reached does not exceed several μm; hence, we used minimum voxel sizes of 4.5 and 5 μm [32]. However, a modern synchrotron radiation facility can provide a resolution of one micron per voxel or less (down to hundreds of nanometers) [102]. Synchrotron radiation facilities also support phase contrast, which might prove useful for imaging biological tissues (including tumor tissue) at high resolution [41,103,104]. However, such facilities exist in limited numbers worldwide; therefore, the development of an imaging protocol that is suitable for widespread laboratory micro-CT scanners may be useful for future cancer biology research.

We did not perform immunohistochemical staining due to the long sample investigation (several months), which could introduce artifacts [105,106]. However, *Apps* et al. performed immunohistochemistry after the micro-CT imaging of samples pretreated with phosphotungstic acid [40], suggesting that micro-CT imaging could be complemented with this technique. Immunohistochemistry is essential for cancer research because of its ability to assess the expression of a wide variety of markers associated with invasion and epithelial-mesenchymal transition [107,108]. Moreover, epigenetic studies combined with micro-CT can be used to analyze DNA methylation in tumors and study the epigenetics of tumor invasion [109]. Zhou et al. recently revealed spatial heterogeneity in esophageal SCC by analyzing the tumor microenvironment and quantifying subclonal expansion [110]. During spatially directed evolution, subclones mostly originated from the tumor center but had a biased clonal expansion to the upper direction of the esophagus. Furthermore, Li et al. analyzed the genotypes and phenotypes of bladder cancer cells in 3D tumor spheroids during collective cancer invasion; cells that formed the finger-like projections had higher MALAT1 RNA expression levels than cells in the outer shell of the spheroids [111]. However, further research is required to determine whether immunohistochemistry and multiomic studies are compatible with micro-CT imaging.

In our study, we did not use automatic segmentation based on artificial intelligence; however, it might improve and accelerate the accurate characterization of cancer progression and metastasis via micro-CT. For instance, Buccardi et al. applied deep learning algorithms to estimate the spatial distribution of fibrotic lesions in a murine model of lung fibrosis [112]. Schoppe et al. used the deep learning solution AIMOS, which automatically segmented organs to localize cancer metastases in mice [113]. However, to the best of our knowledge, this is the first study to describe the routes of tumor spread using 3D visualization, intensity value analysis, and segmentation, and to compare the micro-CT data with the results of histological examination. Compared to the manual semiautomatic segmentation that was performed in this study, the existing deep learning algorithms [114,115] cannot provide a detailed characterization of tumor spread (e.g., to visualize strands and finger-like projections) via preclinical micro-CT.

## 6. Conclusions

In this study, we applied micro-CT imaging to visualize tumor progression in orthotopic esophageal squamous cell carcinoma PDOX tumors, capturing primary tumor lesions, various sites of local invasion, and metastatic lesions. By performing serial scans at different resolutions and generating three-dimensional digital models, we preserved the native tissue architecture, offering insights unattainable with conventional histology alone.

Our analysis revealed a range of invasion patterns during progression of esophageal cancer, such as expansive growth, multicellular streaming, and the formation of strands and finger-like projections, which we visualized in three dimensions. Additionally, we visualized secondary metastatic lesions and analyzed their structural relationships with surrounding tissues. This was particularly significant for peritoneal carcinomatosis, where three-dimensional visualization demonstrated complex mechanisms of collective migration that extend beyond traditional concepts of passive spread. Tumors exploited pre-existing anatomical structures, such as peritoneal folds and ligaments, as scaffolds for collective dissemination. Notably, we confirmed the concept of cyclical tumor dissemination, wherein newly formed metastatic lesions act as active hubs for further systemic or local spread.

Furthermore, we demonstrated that the phenomenon of tumor budding—often interpreted as isolated clusters in 2D histological sections—can frequently be attributed to sectioning artifacts. In 3D reconstructions, these buds were visibly connected to the primary invasion front, challenging traditional histological interpretations. Our findings also provide evidence of adaptive phenotypic plasticity, with tumor cells dynamically adjusting to distinct microenvironmental conditions to optimize proliferation and collective migration.

These insights underscore the transformative potential of micro-CT for capturing the full complexity of tumor invasion and dissemination. This technique not only reveals sophisticated tissue-remodeling strategies that extend beyond enzymatic matrix degradation but also offers a more comprehensive understanding of tumor-tissue interactions. By integrating micro-CT with conventional histology, a multimodal imaging approach emerges, offering a clearer and more accurate depiction of tumor behavior.

Finally, our observations suggest that collectively migrating tumor cells exhibit organized behavior reminiscent of social amoebae and early multicellular organisms, highlighting their adaptive capabilities and deep evolutionary potential. Looking ahead, the broader application of micro-CT in cancer research and diagnostics could refine therapeutic strategies, improve prognostic accuracy, and advance our understanding of the mechanisms driving cancer progression.

## Data Availability

The data that support the findings of this study are openly available in https://zenodo.org/records/13990467 (accessed on 26 March 2025).

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
