# Peer review of "Unveiling Another Dimension: Advanced Visualization of Cancer Invasion and Metastasis via Micro-CT Imaging"

_cancers, 2025, doi:10.3390/cancers17071139_

Round 1
Reviewer 1 Report
Comments and Suggestions for Authors
This study used computed microtomography (micro-CT) to visualize tumor growth, invasion, and metastasis at high resolution, preserving the natural tissue microarchitecture. It also visualize invasion patterns such as tumor budding and several types of tumor strands. Some points should be noted as below,
1) The authors claim that it can enable a detailed study of tumor interactions with surrounding tissues. Can the interaction between immune cells and tumors including budding cells be detected?
2) Tumor buddings typically are defined as the presence of isolated single cancer cells or clusters of up to four cancer cells at the invasive tumour front (Lugli A, et al. Tumour budding in solid cancers.Nat Rev Clin Oncol. 2021.). There are several marked tumor buds in the paper, but in fact, they are not typical in terms of morphology (Fig. 3D, Fig.4D, etc). Additionally, Fig.2A, How can effectively distinguish this arrow-marked area as a tumor bud?
3) One interesting paper proposes that cancer is a pathological ecosystem, and tumor-host interface (invasive front) is the ecological transition zone of cancer, and tumor buddings should be recognized as ecological islands separated from tumor clusters (https://pubmed.ncbi.nlm.nih.gov/37056571/). It is suggested that the authors conduct an appropriate discussion in light of the findings in this research.
4) To be honest, the morphological changes presented by these images, such as tumor budding or tumor invasion, are not as intuitive as HE staining, and may even produce some false impressions. Even cancer cells can often be difficult to distinguish from the surrounding tissue. How can these limitations be addressed to be applied in clinical practice?
Reviewer 2 Report
Comments and Suggestions for Authors
The title “Unveiling another dimension: advanced visualization of cancer invasion and metastasis via micro-CT imaging” is appropriate for this manuscript.
The manuscript discusses a patient-derived orthotopic model of esophageal squamous cell carcinoma and analysis with micro-CT imaging and subsequent histological analysis of iodine-stained specimens. The 3D structure of the tumor and adjacent tissues was reconstructed preserving the native histological architecture and well-known tumor invasion patterns such as budding and different types of tumor filaments were visualized. The authors demonstrated peritoneal tumor dissemination in three dimensions and metastatic colonies formation. Based on the acquired data, possible tumor spread pathways in experimental animals depending on the tissue microenvironment and adaptive evolution of tumor cells in secondary lesions and metastases were also proposed.
Overall the manuscript is well structured, but CT imaging is not highly valued from the point of view of quantitative evaluation. This part should be considered in the text based on the current state of the art.
Please see the comments below.
1) It is strongly recommended that the authors consider expanding the discussion by including the following points: [10.3390/jimaging8040092.]. This suggestion is based on the need to mention in the text that the visualization of cancer and therefore of the invasion and development of metastases is generally done with PET imaging assessment associated with CT and MR. These methods are minimally invasive and translational as they can be applied from in vivo models to the clinic. The experimental evidence presented by the authors should be contextualized and discussed in the discussion section based on gold-standard techniques and methods. Furthermore, in the proposed article, a radiomic workflow that relies on analysis through artificial intelligence tools on mice is also shown for the evaluation of the uptake and biodistribution of new radiopharmaceuticals in xenografted models with tumor cells. Since the article talks about orthotopic animal models and evaluation through morphological CT imaging analysis and since radiomics is widely used in a predictive way on metabolic but additionally morphological imaging, it is strongly recommended, in terms of future perspectives, to mention the possibility of applying a similar workflow to distinguish the targets discussed in the proposed article. This would make the manuscript highly topical and highly applicable to researchers using AI-based approaches for preclinical image analysis.
2) [10.1016/j.pharmthera.2024.108631] The authors are strongly encouraged to use the latest state of the art. For example, in this manuscript the use of orthotopic and metastatic tumor models in preclinical cancer research has been discussed, outlining some of their main advantages and limitations and highlighting considerations associated with the use of orthotopic and metastatic tumor models as part of anticancer drug development. This will help to further justify the authors' choice of the model they used and thus give scientific validity to the study.
3) Figure 11 should be moved to the discussion section or introduction.
3)To make reading the document easier for the reader, an Abbreviations section should be added to the end of the document.
4) Finally, to make the article more coherent, the references should be extended, and English should be improved.
Comments on the Quality of English LanguageEnglish should be improved.
Round 2
Reviewer 2 Report
Comments and Suggestions for Authors
The title “Unveiling another dimension: advanced visualization of cancer invasion and metastasis via micro-CT imaging” is appropriate for this manuscript.
The manuscript discusses a patient-derived orthotopic model of esophageal squamous cell carcinoma and analysis with micro-CT imaging and subsequent histological analysis of iodine-stained specimens. The 3D structure of the tumor and adjacent tissues was reconstructed preserving the native histological architecture and well-known tumor invasion patterns such as budding and different types of tumor filaments were visualized. The authors demonstrated peritoneal tumor dissemination in three dimensions and metastatic colonies formation. Based on the acquired data, possible tumor spread pathways in experimental animals depending on the tissue microenvironment and adaptive evolution of tumor cells in secondary lesions and metastases were also proposed.
Overall the manuscript is well structured, but CT imaging is not highly valued from the point of view of quantitative evaluation. This part should be considered in the text based on the current state of the art.
Please see the comments below.
In answer 1 the authors state that "We focused on micro-CT studies, since in our study, we aimed to track tumor progression and metastases at the level of cell clusters at the highest possible resolution with subsequent histological examination of these regions, while PET imaging assesses cancer progression at the whole organ level by direct measurement of chelator and radiotracer uptake at a given time." Please add this answer and the relevant bibliographic references in lines 695-707, since at the moment the assessment of tumor metabolism and therefore progression is conducted via nuclear medicine approaches, so the scientific choice adopted for a broad-spectrum discussion must be justified. The paper in its current form does not have an immediate translational applicability that could go into the clinic, therefore this limitation must also be underlined in the text. This is not exhaustive.
In comments 3 and 5, it was requested to move a figure in the text and expand the state of the art to support the experimental evidence discussed. The authors did not respond exhaustively, thanking for having correctly reported these two points, but without supporting the suggested requests. This is not exhaustive.
In the comment on the quality of English, which asked for improvements in the language, the authors do not comply with the request, stating the following: “We apologize for the typos and mistakes of our manuscript. These changes did not influence the content and framework of our paper. Thus, we did not list or mark the changes in the revised paper. We hope that the language has been improved.” This is not exhaustive.
Comments on the Quality of English LanguageIn the comment on the quality of English, which asked for improvements in the language, the authors do not comply with the request, stating the following: “We apologize for the typos and mistakes of our manuscript. These changes did not influence the content and framework of our paper. Thus, we did not list or mark the changes in the revised paper. We hope that the language has been improved.” This is not exhaustive.
Round 3
Reviewer 2 Report
Comments and Suggestions for Authors
The authors have partially responded. However, the new parts added in the text are not supported by currently in vogue references and should be expanded as they are essential to describe the current state of the art and the directions cancer imaging is taking for a correct comparison of the limitations and advantages of the proposed method.
Furthermore, it is unclear why the authors responded to:
"Comment 1: In answer 1 the authors state that "We focused on micro-CT studies, since in our study, we aimed to track tumor progression and metastases at the level of cell clusters at the highest possible resolution with subsequent histological examination of these regions, while PET imaging assesses cancer progression at the whole organ level by direct measurement of chelator and radiotracer uptake at a given time." Please add this answer and the relevant bibliographic references in lines 695-707, since at the moment the assessment of tumor metabolism and therefore progression is conducted via nuclear medicine approaches, so the scientific choice adopted for a broad-spectrum discussion must be justified The paper in its current form does not have an immediate translational applicability that could go into the clinic, therefore this limitation must also be underlined in the text."
in this way:
"Response 1: We thank the reviewer for pointing out the necessity of implementing our answer in the manuscript text and adding the relevant bibliographic references. In line with the reviewer's comment, we have added a paragraph in the Discussion section, part “2.6 Limitations of the study”, lines 841-852. "
The authors added in manuscript this:
"We chose micro-CT as the main method for our study because it allows to track tumor progression and metastases at the level of cell clusters with the highest possible resolution, followed by histological examination of these regions [85,31,58,59] While PET imaging assesses cancer progression at the whole-organ level through direct measurement of chelator and radiotracer uptake at a given time [86,87], its resolution could not provide the detailed histological picture than information on morphological properties of tissues, but has faster translational applicability [86,88,87]. We hypothesize that future advances may enable the combination of in-vivo PET imaging with ex-vivo micro-CT imaging enhanced by contrast agents and correlated with histology in areas of interest [58,86]. This integration of three complementary techniques could provide a comprehensive picture of tumor structure, its interactions with surrounding tissues, and detailed metabolic characteristics of primary or metastatic tumors."
Why hasn't the highly appropriate reference for this part been included in this part and which shows a highly innovative approach of minimally invasive biodistribution based on radiomics? By not including this reference [10.3390/jimaging8040092.], the manuscript obscures recent work in this field of molecular imaging. The authors are strongly requested to consider this suggestion as it would make the discussion more current and based on recent state of the art.
The inconsistency arises from the fact that it was initially suggested to take this comment into consideration:
"It is strongly recommended that the authors consider expanding the discussion by including the following points: [10.3390/jimaging8040092.]. This suggestion is based on the need to mention in the text that the visualization of cancer and therefore of the invasion and development of metastases is generally done with PET imaging assessment associated with CT and MR. These methods are minimally invasive and translational as they can be applied from in vivo models to the clinic the discussion section based on gold-standard techniques and methods. Furthermore, in the proposed article, a radiomic workflow that relies on analysis through artificial intelligence tools on mice is also shown for the evaluation of the uptake and biodistribution of new radiopharmaceuticals in xenografted models with tumor cells. Since the article talks about orthotopic animal models and evaluation through morphological CT imaging analysis and since radiomics is widely used in a predictive way on metabolic but additionally morphological imaging, it is strongly recommended, in terms of future perspectives, to mention the possibility of applying a similar workflow to distinguish the targets discussed in the proposed article. This would make the manuscript highly topical and highly applicable to researchers using AI-based approaches for preclinical image analysis."
Although the authors have included a part in support of this initial comment, it is not exhaustive and there is no reference to this highly significant radiopharmaceutical biodistribution work in the field of oncology imaging and minimally invasive analysis techniques.
Please revise this part accordingly and add the correct parts and correct references.
